# Epidemiology and care management of Paroxysmal Nocturnal Hemoglobinuria (PNH) in a real-world setting in France: Description from the French National Hospitalization Database

Régis Peffault de Latour[1], Sophie Pibre[2], Julien Vercruyssen[2]*,
Marie Laetitia Desjuzeur-Bellais[2], David Pau[2], Stève Bénard[3], Louis Chillotti[3],
Nathalie Grandfils[2]

1 Saint-Louis Hospital, APHP, Paris, France, 2 Roche S.A.S., Boulogne-Billancourt, France, 3 Stève consultants, a Cytel company, Oullins-Pierre-Bénite, France

* julien.vercruyssen@roche.com

## Abstract

### Introduction

Paroxysmal nocturnal hemoglobinuria (PNH) is a rare, acquired stem-cell disease causing anemia, aplasia and thromboses, affecting ~1/80,000 patients in France. Treatment mainly involves C5 inhibitors (C5i) and iterative red-blood-cell transfusions. Recent real-world French data on PNH epidemiology and management are scarce in the era of complement inhibitors. This study described PNH epidemiology, patients' characteristics, and C5i use.

### Methods

This was a non-interventional claims study using secondary data from the French national hospital database. Patients with a hospital diagnosis of PNH (ICD-10 code D59.5) over 2018–2022 were selected, after exclusion of those with other C5i-treated diseases. A subgroup of C5i initiators was followed from initiation to last hospitalization, to describe C5i treatment patterns.

### Results

The study included 897 patients: 496 (55.3%) receiving a C5i and 218 (24.3%) initiating treatment over 2018–2022. In 2022, 725 PNH patients were identified (prevalence ~1/94,000). Mean (SD) age was 52.1 (21.1) and 45.7 (19.4) years overall and among C5i initiators, respectively. C5i initiators had 20.9 (6.9) dispensations annually. Most patients initiated with eculizumab (n = 196, 89.9%), and almost half of them (n = 80, 40.8%) switched to ravulizumab.

**Data availability statement:** The data that support the findings of this study are available from French Health Insurance Database (SNDS), but restrictions apply to the availability of these data, which can only be accessed as part of MR-006 expedited regulatory process and so are not publicly available. Only authorized personnel can access these data, on preidentified secured platforms, after the submission of a MR-006 dossier to the French data protection agency (CNIL – see Research not involving human participants). Additional information on MR-006 process and data access are available on CNIL website (https://www.cnil.fr/sites/cnil/files/atoms/files/mr-006_methodologie_de_reference-traitements_de_donnees_du_snds_et_des_rpu_interet_legitime.pdf) and Health Data Hub (HDH) websites (https://www.health-data-hub.fr/). For further information, please contact the HDH using the following link (https://www.health-data-hub.fr/contact), or contact Louis Chillotti (lchillotti@steve-consultants.com).

**Funding:** The study was funded by Roche France. Funds were allocated to stève consultants, a Cytel company, under research contract with Roche France.

**Competing interests:** RPL received honoraria for participation in the development and validation of the study and its results. SP, JV, MLD, DP, NG are employees of Roche France and did not receive study-specific compensations. SB, LC are employees of stève consultants, a Cytel company, and did not receive study-specific compensation. This does not alter authors' adherence to PLOS ONE policies on sharing data and materials. Aside from this study, RPL received honoraria from Novartis, Amgen, Jazz pharmaceuticals, MSD, Sanofi, Pfizer, Swedish orphan biovitrum, Alexion, Roche.

**Abbreviations:** ATC, Anatomical Therapeutic and Chemical classification; CASD, Secured data center (Centre d'accès sécurisé aux données); CCAM, Classification commune des actes médicaux (French procedure classification); CD, Cluster of differentiation; CNIL,

## Conclusion

PNH prevalence was ~1/94,000. Most patients initiated eculizumab, with 40.8% switching to ravulizumab.

## Introduction

Paroxysmal nocturnal hemoglobinuria (PNH) is an acquired clonal hematopoietic stem cell disease characterized by corpuscular hemolytic anemia, bone marrow aplasia and frequent thrombosis and with an estimated prevalence of 1/80,000 in France [1]. The disease is caused by somatic mutations in the PIG-A gene (Xp22.1), which codes for a protein required for the biosynthesis of glycosyl-phosphatidylinositol (GPI). The mutation occurs in one or more hematopoietic stem cell(s) and results in a total – or partial – deficiency of all the proteins which normally attach to the surface of the cell membrane via the GPI anchor molecule (including CD55 and CD59, which regulate complement-related hemolysis). PNH can affect people of any age but is mostly encountered among young adults. Clinical manifestations are variable and may include hemolytic anemia with a notable hemoglobinuria observable in the morning, medium vessel thromboses, and hematopoiesis deficiency, which may lead to pancytopenia. The hemolysis brings several complications, with anemia being the most frequent (pallor, fatigue, and shortness of breath), jaundice, and, in some cases, renal failure. Depending on their location, thrombosis (which affects 30–40% of patients without treatment) can lead to abdominal pain, hepatomegaly, ascites, and headache [1,2].

Currently, the only curative treatment available for PNH is hematopoietic stem cell transplantation, which remains rarely used due to the lack of donors, compatibility concerns, and potential severe complications. It is usually proposed to patients with concomitant severe aplastic anemia or other causes of bone marrow failure, addressing both diseases [3–5]. Symptomatic treatment is based on iterative red blood cell transfusions, anticoagulants, corticosteroids, or treatment of associated aplasia. Eculizumab, an intravenous monoclonal antibody targeting the complement component C5 (i.e., C5 inhibitor), was the first etiologic treatment for PNH, made available in 2007 in France. More recently, alternatives were developed with ravulizumab, another intravenous C5 inhibitor, and pegcetacoplan, a subcutaneous therapy targeting the complement component C3 – indicated after C5 inhibitor failure or intolerance – available since 2020 and 2022, respectively [2,6–10]. Finally, in August 2024, crovalimab, a new C5 inhibitor received a European market access authorization for the treatment of PNH. Unlike current C5 inhibitors, which are intravenous, crovalimab will be dispensed as a monthly subcutaneous injection.

Real-world data on PNH epidemiology and general care pathways in France are scarce and could be used by Health authorities to assess PNH's impact on public health in the era of these new therapies. The main objectives of this study were to describe PNH epidemiology and patients' characteristics, the therapeutic pathways of patients treated with C5 inhibitors), and other related therapies such as C3 inhibitor (which was only available in the last months of the study period), in a real-world setting in France.

French data protection agency (Commission national de l'informatique et des libertés); GPI, Glycosyl-phosphatidylinositol; ICD-10, International Classification of Diseases – 10th revision; IQR, Interquartile range; ISPE, International society of pharmacoepidemiology; PNH, Paroxysmal nocturnal hemoglobin-uria; PMSI, French hospitalization database (Programme de médicalisation des systèmes d'information); SD, Standard deviation

## Materials and methods

### Design and patient selection

This was a non-interventional, national, claims study using secondary data from the French national hospital database (PMSI). The PMSI is the French exhaustive hospital claims database, covering every stay of patients covered by national health insurance (99% of French residents) and containing patients' demographic, clinical, and therapeutic data, and in-hospital deaths for all hospitalizations in public or private hospitals (~65 million individuals). Main data available in PMSI are the hospital center (identifier, geographical area, setting), admission and discharge date, diagnoses reported during hospitalization using international classification of diseases – 10th revision (ICD-10), diagnosis-related group, medical procedures using the French procedure classification (CCAM), expensive drugs and medical devices administered on top of the stay ("liste en sus"), stays in special units (resuscitation, intensive care, emergency room, neonatology or palliative care), hospital specialist visits [11].

All of the patients with at least one diagnosis of PNH based on the code D59.5, with or without at least one dispensation of C5 inhibitor between January 1, 2018, and December 31, 2022 (selection period), were selected (S1 Fig in S1 File).

Among these patients, those presenting another disease potentially treated with a C5 inhibitor (i.e., hemolytic-uremic syndrome, myasthenia gravis, neuromyelitis optica, or other demyelinating diseases of central nervous system) were excluded. This exclusion criterion was validated using an iterative approach. Patients were first grouped according to the number of exclusion-disease markers identified (0, 1, 2, or ≥3 ICD-10 diagnoses). A progressively more restrictive exclusion strategy was then applied to estimate how many patients would remain eligible, and these estimates were compared with figures from literature and expert opinion. This process ultimately supported the exclusion of all patients presenting with at least one marker of an exclusion disease. Study cohort inclusion date corresponded to the first marker of PNH (hospitalization or C5 inhibitor dispensation) over the selection period.

A subgroup of patients initiating a C5 inhibitor during the selection period was analyzed to describe C5 inhibitor therapeutic pathways (anatomical therapeutic and chemical (ATC) codes L04AA25 for eculizumab and L04AA43 for ravulizumab at the time covered by this study). Of note, eculizumab and ravulizumab can only be dispensed during hospital stays in France and are part of the "liste en sus" (S1 Table in S1 File).

Patients' characteristics were described at inclusion and, for the subgroup of patients defined above, at C5 inhibitor initiation. An historical period since January 1, 2014 was defined to describe patients' medical and treatment history including previous use of transfusions. Patients initiating a C5 inhibitor were followed from treatment initiation until the last hospital stay available.

Codes and algorithms for patient selection and definition of variables were derived from validated algorithms and are available in Supplementary material (S2, S3, S4 Tables in S1 File).

## Outcomes and variables

The sociodemographic characteristics and the medical history were described at inclusion and at C5 inhibitor initiation. Sociodemographic characteristics gathered age, gender, and administrative area of residence. Clinically significant comorbidities investigated over medical history included diabetes, hepatic failure, myocardial infarction, valvular diseases, hypertension, heart failure, pulmonary embolism, and stroke.

Among patients initiating a C5 inhibitor, therapeutic pathways were described through the number of stays with a C5 inhibitor dispensation and the type of C5 inhibitor used, over the follow-up period.

## Statistical analyses

Continuous data were summarized as mean (standard deviation – SD) or median (interquartile range – IQR) and categorical variables were presented as number and frequencies. Analyses were carried out using SAS® version 9.4 (SAS Institute Inc. Cary, NC, USA).

C5 inhibitor therapeutic pathways were described using Sunburst visualization, each radius identifies the proportion of patients involved in a treatment pathway. The central ring represents the first dispensation, and the rings moving away from the central ring represent the subsequent treatments.

## Ethics

This study was designed according to the International Society for Pharmacoepidemiology (ISPE) guidelines and applicable regulatory requirements, including the French Data Protection Agency (CNIL) act n°2018−257 on regulatory requirements and authorization for processing PMSI data (MR-006) [12–14]. As per MR-006 process, data were accessed on a secured platform by trained and authorized professionals from stève consultants. No personal data was extracted from the platform. Roche S.A.S only accessed aggregated data.

Finally, a disease area expert, focused on aplastic anemia and PNH, was involved from the protocol's development to ensure scientific and methodological robustness, as well as the clinical relevance of the study and its results.

# Results

## Main characteristics

Among the 917 patients presenting at least one hospitalization with a PNH diagnosis between 2018 and 2022, associated or not with a dispensation of C5 inhibitors, 20 (2.2%) patients presented at least one diagnosis of exclusion diseases, leading to 897 (97.8%) patients included in the cohort. More than half were already treated with a C5 inhibitor (n = 496, 55.3%), and 218 (24.3%) initiated their treatment during the selection period (Fig 1).

Over the study period, the annual number of patients with PNH increased from 581 in 2018–725 in 2022, corresponding to an average prevalence of ~1/94,000. During this period, around 100 incident PNH patients were identified each year on average, with a slight decrease in 2020 (83 patients), followed by a peak in 2021 (129 patients). Overall, around 60% of the patients received at least one dispensation of a C5 inhibitor during the study period. In 2022, among the 100 new patients identified with PNH, 28 patients initiated a C5 inhibitor over the year (Fig 2).

Among the 897 patients of the cohort, the mean (SD) age at inclusion was 52.4 (21.1) years, with 35.0% (n = 314) of patients aged 65 years or older. Slightly more than half (n = 461, 51.4%) of the patients were women. The most frequent comorbidities were cardiac arrhythmia (n = 84, 9.4%), hepatic failure (n = 78, 8.7%) and diabetes (n = 78, 8.7%). Patients initiating a C5 inhibitor tended to be younger, with a mean (SD) age of 45.7 (19.4) years, and to have less frequent comorbidities, with 5.5% (n = 12) of patients diagnosed with cardiac arrhythmia and 6.9% (n = 15) with diabetes (Table 1).

## Treatment modalities

Among the 218 patients initiating a C5 inhibitor, the mean follow-up duration was 25.6 (17.3) months. The first dispensation was almost exclusively done in a university hospital (90.3%), or in other public hospitals (6.9%). The mean (SD)

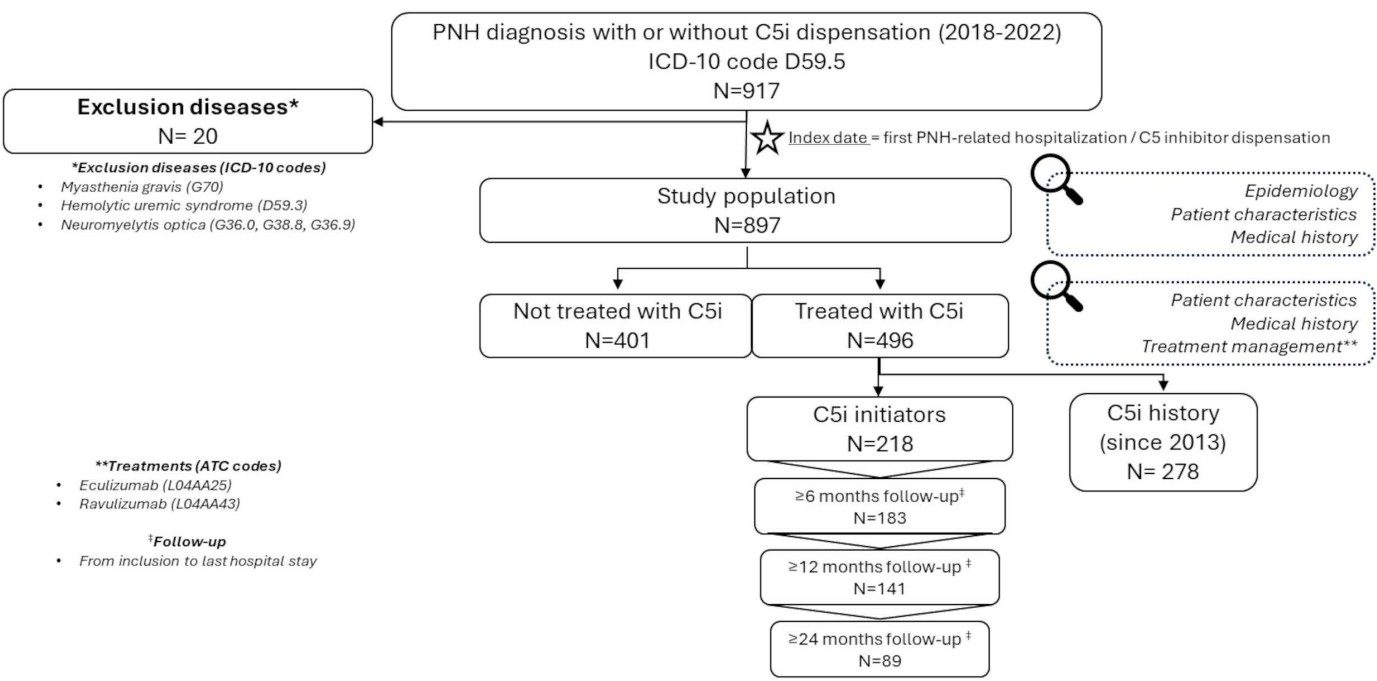

**Fig 1. Patient disposition and subgroups of interest.**

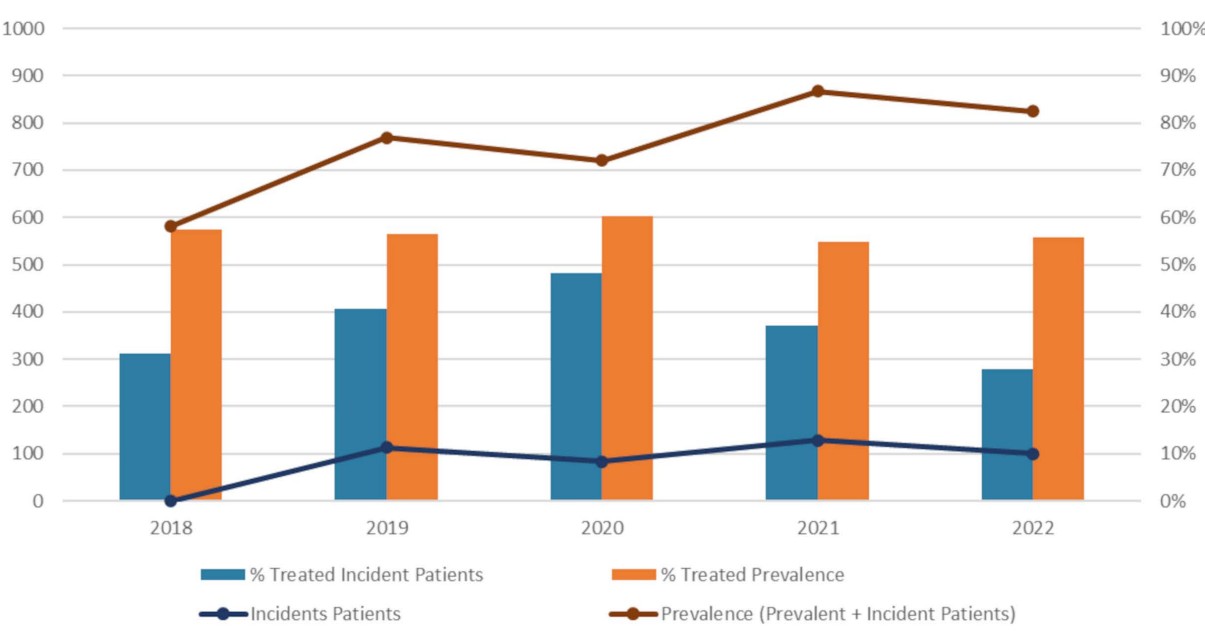

**Fig 2. Number of patients with PNH during study period and proportion of patients treated with C5 inhibitors.**

**Table 1. Main sociodemographic and clinical characteristics.**

| | Patients initiating a C5 inhibitor N = 218 | Study cohort N = 897 |
|---|---|---|
| Age | | |
| Mean (SD) | 45.7 (19.4) | 52.4 (21.1) |
| Median (IQR) | 44.5 (27.0 - 61.0) | 53.0 (34.0 - 70.0) |
| Sex | | |
| Female, n (%) | 100 (45.9%) | 461 (51.4%) |
| Male, n (%) | 118 (54.1%) | 436 (48.6%) |
| Comorbidities of interest | | |
| Arrythmia, n (%) | 12 (5.5%) | 84 (9.4%) |
| Other thrombotic diseases, n (%) | 18 (8.3%) | 82 (9.1%) |
| Diabetes, n (%) | 15 (6.9%) | 78 (8.7%) |
| Hepatic failure, n (%) | 15 (6.9%) | 78 (8.7%) |
| Renal failure, n (%) | <11 (<5.0%) | 63 (7.0%) |
| Heart failure, n (%) | 15 (6.9%) | 56 (6.2%) |
| Myocardial infarction, n (%) | <11 (<5.0%) | 54 (6.0%) |
| Stroke, n (%) | <11 (<5.0%) | 35 (3.9%) |
| Pulmonary embolism, n (%) | <11 (<5.0%) | 18 (2.0%) |

Other thrombotic diseases: atherosclerosis, peripheral vascular diseases, phlebitis and thrombophlebitis.

IQR: inter-quartile range, SD: standard deviation.

As per MR-006 process, results in samples of less than 11 patients cannot be displayed.

annual number of hospitalizations with a C5 inhibitor dispensation was 20.9 (6.9), which leads to an average frequency of 1 injection each ~17 days. Among them, 196 (89.9%) patients initiated with eculizumab and only 22 (10.1%) patients initiated with ravulizumab (available since March 2022 in France). Considering patients initiating with eculizumab, 80 (40.8%) switched to ravulizumab during their follow-up, and 8 of them ultimately re-switched to eculizumab. Switches were almost exclusively done in university hospitals (86.3%), or in other public hospitals (10.1%). No patient initiating with ravulizumab switched to eculizumab during their follow-up (Fig 3). Finally, fewer than 10 patients were treated with pegcetacoplan, as it was only available in the last months of the study, as early access.

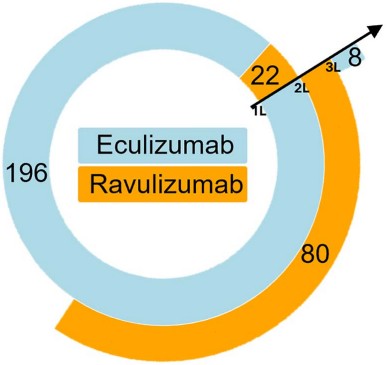

**Fig 3. Treatment patterns of patients initiating a C5 inhibitor (n = 218) – Sunburst representation.**

Sunburst are displayed from the inside to the outside. The first disc represents first line therapy, with 196 patients treated with eculizumab and 22 patients with ravulizumab. The second disc represents patients with first line therapy who switched to a second line. Here, 80 patients treated with eculizumab switched to ravulizumab. Finally, of these 80 patients, 8 switched back to eculizumab.

## Discussion

To the best of our knowledge, this is the first French study to provide real-world data on the epidemiology and hospital management of PNH in France, using data from French national hospital database.

With this study, 725 patients with PNH were identified in 2022, representing a prevalence of 1/94,000, while it was estimated around 1/80,000 by Orphanet and the French rare disease network, and even 1/70,000 according to the latest national plan for PNH diagnosis and treatment dated 2023 [1,15,16]. This highlights a potential underestimation of PNH disease prevalence in our study. Hence, some patients with non-severe PNH might not require etiologic treatments, nor hospitalizations, or might even not be diagnosed, as the disease and its clinical manifestations are directly linked to the proportion of circulating PNH clones. It is also possible that some patients with PNH might have been reported with a non-specific code (e.g., D59.9 Acquired hemolytic anemia, unspecified), ultimately reported as PNH in 2023, after the study period; potentially highlighting a delay in establishing a positive diagnosis of this rare disease. In a recent study from United Kingdom, the prevalence of blood samples with PNH clones was estimated around 3.81/100,000 (or 1/26,250 – corresponding to 2,400 patients), but the prevalence of clinical PNH could not be assessed, emphasizing the difficulty of assessing the current burden of PNH [17]. Overall, the total number of patients increased by around 40 annually, with 100 patients being newly diagnosed each year. The difference of 60 patients is due to loss of follow-up or more likely patients who died outside of the hospital, tending to emphasize the severity of the disease.

In this study, patients were aged 52.4 years in mean, with 51.4% being women. Patients initiating a C5 inhibitor had a mean age of 45.7 years, and 54.1% were female. According to the international PNH registry, gathering data from more 4,000 patients, 53.0% of patients are women, in line with our study. When considering patients initiating a C5 inhibitor in the registry, mean ages were also relatively similar, with 45.1 years. It is of note that this age is the age at inclusion in the study and not the mean age at disease onset, which was 39.3 years [18].

In our study, we observed that around 45% of PNH patients did not receive any C5 inhibitor, which was in line with the current practice according to clinical expert opinion. Treatment decisions in PNH rely almost exclusively on symptoms and biological data (e.g., hemolysis, thrombosis, dysphagia, pain, central anemia), which cannot be directly assessed using PMSI data. Conversely, PMSI exhaustively captures reimbursed treatments for eculizumab and ravulizumab, making the risk of non-identification of treated patients almost null. Consequently, aside from a limited risk of miscoding (further minimized by the use of exclusion diagnoses), untreated patients identified in the database are expected to present with subclinical PNH and low-level hemolysis. Such patients typically benefit from active surveillance rather than immediate treatment initiation.

Over the study period (2018–2022), only two etiologic treatments were available (eculizumab and ravulizumab). Peg-cetacoplan was only available over the latest months and was identified in less than 10 patients. More than half of PNH patients were treated with a C5 inhibitor in this study. The most frequent C5 inhibitor was eculizumab, representing 90% of C5 inhibitor initiations, while the remaining 10% were attributable to ravulizumab. Ultimately, 40.8% of eculizumab initiators switched to ravulizumab, which is in line with the two current indications of ravulizumab in France. Hence, three profiles of patients could be identified. The first profile is that of C5 inhibitor (eculizumab) naive patients with PNH requiring systemic treatment, and diagnosed after ravulizumab became available, representing the 10% of patients initiating with ravulizumab. The second profile is that of patients initiating with eculizumab and switching to ravulizumab after showing a clinical stability of the disease for at least 6 months, representing the 40.8% of patients switching to ravulizumab. A last patient profile consists of patients treated with eculizumab and not switching to ravulizumab. For this last group of patients

(59.2% of eculizumab initiators), several reasons could be considered, such as a recent initiation of eculizumab, a switch refusal despite clinical stability, or the absence of clinical stability [19]. Of note that no clinical data are available in the PMSI making it difficult to precisely assess disease severity. In addition, as ravulizumab was only available during the last year covered by the study, the proportion of patients initiating with or switching to ravulizumab is expected to increase in the upcoming years. Real-world data on the use of ravulizumab use is scarce – mainly case reports – and usually focus on effectiveness and tolerance rather than describing the reasons motivating the switch [20–22].

Finally, our study identified a small number of patients who switched back from ravulizumab to eculizumab. This profile, while rare in our study, raises the question on the reason to switch back. Several plausible hypotheses can be proposed. Ravulizumab was initially indicated for patients with a stable disease receiving eculizumab, mainly to reduce injection frequency. In the event of a recent complication or a severe hemolytic crisis, clinicians may consider the need for closer clinical follow-up, including more frequent dosing, and therefore opt for a switch back to eculizumab. This may also apply to patients with specific clinical presentations or comorbidities requiring intensified monitoring. In addition, patients experiencing adverse events under ravulizumab may be considered for a switch to eculizumab.

One of the main limitations in the present study is due to the nature of PMSI itself. As it consists only in hospital data, outpatient information is not available. This could lead to a potential underestimation of PNH cases solely managed outside of hospitals, as described earlier, but also have an impact on outcome description in case of outpatient deaths (not identifiable in the PMSI). This limitation was managed by censoring patients at their latest hospital stay, whatever the cause, using the maximum follow-up available without underestimating the outcomes due to unidentified deaths. Considering comorbidities of interest, PMSI offers only a robust approximation for describing severe forms, which could have a significant impact on patient management and be coded as such during the stay. This implies that the rates of some comorbidities described in this study may be underestimated. Diseases such as myocardial infarction, stroke, or pulmonary embolism, which are almost exclusively managed in hospital are expected to be more comprehensively identified. Finally, PMSI contains limited clinical and biological data, notably severity, clone size, classification, or tolerance, which are key components of PNH management. For the reasons depicted above, this study only aimed to describe PNH epidemiology and general hospital management, with no considerations on treatment response or comparative effectiveness.

It is of note that for the year 2020 (and 2021 to a lesser extent), the COVID-19 global pandemic strongly impacted the overall healthcare services in France, with the deprogramming of multiple hospitalizations in order to limit hospital overload. In this study, this translated into a decrease in the number of newly identified patients in 2020, followed by a peak in 2021. However, the severity and general management of PNH tended to limit the impact of COVID-19 on PNH patients' pathways and treatment.

Despite limitations and biases depicted above, PMSI is a robust tool for the characterization of PNH epidemiology and management in France for its hospital part. Additional studies on both inpatient and outpatient databases (e.g., SNDS in France) should be carried out in order to exhaustively describe PNH clinical and economic burden. Furthermore, this study covers the initial years of ravulizumab availability and the first months of pegcetacoplan availability in France. Combined with the more recent introduction of crovalimab, this highlights the need for updated data to capture evolving treatment patterns in PNH.

## Conclusion

This study brings updated data on the epidemiology and management of PNH in French hospitals using the exhaustive hospital claims database. More than 700 PNH patients were identified in 2022, with 100 newly diagnosed patients each year, leading to a prevalence of ~1/94,000 persons in France. More than 40% of patients with PNH are not treated with a C5 inhibitor, while they remain the main etiologic treatments available. Eculizumab is the most prescribed C5 inhibitor, and almost half of those who initiate it end up switching to ravulizumab.

**Research involving human participants**

This study which required processing of personal data for the purpose of research in the field of health falls under the French Data Protection Act, as described in The French Data Protection Agency (Commission Nationale de l'Informatique et des Libertés – CNIL) opinion n°2018−257 of 7 June 2018 on regulatory requirements and governance for processing PMSI data under expedited regulatory process (MR-006). PMSI data were accessed between September 2023 and March 2025.

This study was conducted in full compliance with relevant guidelines – including the Declaration of Helsinki, as well as the ICMJE guidelines – and French regulations. This study was conducted using anonymized data extracted from an existing database (PMSI); thus, according to French regulations and MR-006 process, there is no need for an informed consent from patients nor for an approval of an Ethic committee or an Institutional Review Board. Patients retain the right to withdraw or object by submitting an explicit request to their local insurance branch (CPAM), in accordance with article 111 of the application decree n° 2019−536 of 29 May 2019, under the Data Protection Act. Detailed instructions for this process are available at: https://www.health-data-hub.fr/politique-de-confidentialite. A collective information note has been added to both stève consultants and Roche websites for this purpose.

## Supporting information

**S1 File. The online version contains supplementary materials.**
(DOCX)

## Acknowledgments

Access to some confidential data, on which is based this work, has been made possible within a secure environment offered by CASD – *Centre d'accès sécurisé aux données* (Ref. 10.34724/CASD).

## Author contributions

**Conceptualization:** Régis Peffault de Latour, Sophie Pibre, Julien Vercruyssen, David Pau, Stève Bénard, Louis Chillotti, Nathalie Grandfils.

**Formal analysis:** Marie Laetitia Desjuzeur-Bellais, David Pau, Stève Bénard, Louis Chillotti.

**Investigation:** Sophie Pibre, Marie Laetitia Desjuzeur-Bellais, David Pau, Louis Chillotti, Nathalie Grandfils.

**Methodology:** Régis Peffault de Latour, Julien Vercruyssen, David Pau, Stève Bénard, Louis Chillotti, Nathalie Grandfils.

**Project administration:** Julien Vercruyssen, Marie Laetitia Desjuzeur-Bellais, Stève Bénard, Nathalie Grandfils.

**Resources:** Sophie Pibre, Nathalie Grandfils.

**Supervision:** Nathalie Grandfils.

**Validation:** Régis Peffault de Latour, Sophie Pibre, Julien Vercruyssen, Marie Laetitia Desjuzeur-Bellais, David Pau, Stève Bénard, Louis Chillotti.

**Visualization:** Nathalie Grandfils.

**Writing – original draft:** Régis Peffault de Latour, Louis Chillotti.

**Writing – review & editing:** Régis Peffault de Latour, Sophie Pibre, Julien Vercruyssen, Marie Laetitia Desjuzeur-Bellais, David Pau, Stève Bénard, Nathalie Grandfils.

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
