## [Decision Letter · Decision Letter 0]

30 Dec 2025

Dear Dr. Chillotti,

 The manuscript has been evaluated by three reviewers, and their comments are available below.The reviewers have raised a number of concerns that need attention. In particular, they request additional information on methodological aspects of the study, and additional data analysis.Could you please revise the manuscript to carefully address the concerns raised? Please submit your revised manuscript by Feb 13 2026 11:59PM. If you will need more time than this to complete your revisions, please reply to this message or contact the journal office at ?>plosone@plos.org. . . . A letter that responds to each point raised by the academic editor and reviewer(s). You should upload this letter as a separate file labeled 'Response to Reviewers'.A marked-up copy of your manuscript that highlights changes made to the original version. You should upload this as a separate file labeled 'Revised Manuscript with Track Changes'.An unmarked version of your revised paper without tracked changes. You should upload this as a separate file labeled 'Manuscript'.

We look forward to receiving your revised manuscript.

Kind regards,

Helen Howard

Staff Editor

PLOS One

Journal Requirements:

https://journals.plos.org/plosone/s/file?id=ba62/PLOSOne_formatting_sample_title_authors_affiliations.pdf....

“This study was funded by Roche SAS”

3. We noted in your submission details that a portion of your manuscript may have been presented or published elsewhere. [The results of this study were presented at the EHA 2024 congress] Please clarify whether this [conference proceeding or publication] was peer-reviewed and formally published. If this work was previously peer-reviewed and published, in the cover letter please provide the reason that this work does not constitute dual publication and should be included in the current manuscript.

“RPL received honoraria from Novartis, Amgen, Jazz pharmaceuticals, MSD, Sanofi, Pfizer, Swedish orphan biovitrum, Alexion, Roche

SP, JV, MDB , DP and NG are employees of Roche SAS

SB and LC are employees of stève consultants – a Cytel company, which has a research contract with Roche SAS”

We note that one or more of the authors are employed by a commercial company

5. We note that you have indicated that there are restrictions to data sharing for this study. For studies involving human research participant data or other sensitive data, we encourage authors to share de-identified or anonymized data. However, when data cannot be publicly shared for ethical reasons, we allow authors to make their data sets available upon request. For information on unacceptable data access restrictions, please see http://journals.plos.org/plosone/s/data-availability#loc-unacceptable-data-access-restrictions.

“This study was funded by Roche SAS. Access to some confidential data, on which is based this work, has been made possible within a secure environment offered by CASD – *Centre d’accès sécurisé aux données* (Ref. 10.34724/CASD).”(Ref. 10.34724/CASD).”(Ref. 10.34724/CASD).”(Ref. 10.34724/CASD).”

Please remove any funding-related text from the manuscript and let us know how you would like to

“This study was funded by Roche SAS”

Reviewers' comments:

Reviewer's Responses to Questions

**Comments to the Author**

1. Is the manuscript technically sound, and do the data support the conclusions?

Reviewer #1: Yes

Reviewer #2: Yes

Reviewer #3: Yes

2. Has the statistical analysis been performed appropriately and rigorously?

Reviewer #1: Yes

Reviewer #2: Yes

Reviewer #3: Yes

3. Have the authors made all data underlying the findings in their manuscript fully available?

Reviewer #1: Yes

Reviewer #2: Yes

Reviewer #3: Yes

4. Is the manuscript presented in an intelligible fashion and written in standard English?

Reviewer #1: Yes

Reviewer #2: Yes

Reviewer #3: Yes

Reviewer #1: I want to congratulate the authors on submitting this well written manuscript

I have the following queries and suggestions-

1) The data collected is from 2018-2022. It would be a good idea to include data till the year 2024-25.

2) What was the treatment response on follow up. Please include the details.

3) To make the data more meaningful, it would be nice to compare the responses and outcomes of eculizumab vs ravalizumab and pegcetacoplan.

4) What were the various reasons for patients not opting for C5i therapy as these are nearly 50% patients.

5) What were the reasons for patients switching back to eculizumab from ravalizumab? Kindly elaborate.

Reviewer #2: This is a valuable article addressing the prevalence of PNH in France and current treatment strategies.

As a minor point, in line 71, the expression “unlike the three aforementioned drugs, which are intravenous” appears to imply that pegcetacoplan is administered intravenously; however, pegcetacoplan is administered subcutaneously. Clarification or rephrasing would improve accuracy.

Reviewer #3: This manuscript presents a national, real-world analysis of PNH epidemiology and hospital-based management in France using PMSI data from 2018–2022. The use of an exhaustive national hospital database is a clear strength and the topic is timely given the evolving complement inhibitor landscape. The manuscript is generally well written and methodologically appropriate for a descriptive claims-based study.

There are some key points that may benefit from revision to strength interpretation.

1. Case identification relies solely on ICD-10 code D59.5 during hospitalisation. Although difficulties in epidemiological prevalence are well acknowledged. The manuscript would benefit from clearer acknowledgement on that this is PMSI-estimated prevalence. Please also clarify whether any additional criteria (e.g. repeated coding, associated procedures or treatments) were considered to increase diagnostic specificity.

2. There is limited data on PNH clone size and classification of PNH, either in the diagnostic setting or in those patients that received therapy - comment should be made if this was not available.

3. It would strengthen the manuscript if there is any common symptoms data available in the group that received complement inhibitory therapy vs. those untreated.

4. Switching from eculizumab to ravulizumab is well described but interpretation is limited by the absence of clinical data. Is there any observational data?

5. The sunburst figure is informative but would benefit from additional explanation or a complementary tabular summary.

In summary, this is a valuable descriptive study providing updated national data on hospital-managed PNH in France. With changes listed, the manuscript would be suitable for publication.

.

Reviewer #1: **Yes:** Akanksha GargAkanksha GargAkanksha GargAkanksha Garg

Reviewer #2: No

Reviewer #3: No

---

## [Author Response · Author response to Decision Letter 1]

27 Feb 2026

We would like to thank the reviewers and editors for their valuable feedback. We attached a point-by-point response to address each comments

---

## [Decision Letter · Decision Letter 1]

30 Mar 2026

Epidemiology and care management of Paroxysmal Nocturnal Hemoglobinuria (PNH) in a Real-World Setting in France: Description from the French National Hospitalization Database

PONE-D-25-54111R1

Dear Dr. Chillotti,

We’re pleased to inform you that your manuscript has been judged scientifically suitable for publication and will be formally accepted for publication once it meets all outstanding technical requirements.

Kind regards,

Marianne Clemence

Staff Editor

PLOS One

Additional Editor Comments (optional):

Reviewers' comments:

Reviewer's Responses to Questions

**Comments to the Author**

Reviewer #1: All comments have been addressed

2. Is the manuscript technically sound, and do the data support the conclusions?

Reviewer #1: Yes

3. Has the statistical analysis been performed appropriately and rigorously?

Reviewer #1: Yes

4. Have the authors made all data underlying the findings in their manuscript fully available?

Reviewer #1: Yes

5. Is the manuscript presented in an intelligible fashion and written in standard English?

Reviewer #1: Yes

Reviewer #1: The authors have addressed all the queries and made relevant changes in the manuscript. The manuscript can be accepted for publication.

.

Reviewer #1: **Yes:** Akanksha GargAkanksha GargAkanksha GargAkanksha Garg

---

## [Editor Report · Acceptance letter]

PONE-D-25-54111R1

PLOS One

Dear Dr. Chillotti,

I'm pleased to inform you that your manuscript has been deemed suitable for publication in PLOS One. Congratulations! Your manuscript is now being handed over to our production team.

Kind regards,

on behalf of

Dr Marianne Clemence

Staff Editor

PLOS One